# OpenReview forum: "RedCodeAgent: Automatic Red-teaming Agent against Diverse Code Agents"
_ICLR.cc/2026/Conference — ICLR 2026 Poster_

### Official Review · Reviewer_WdER · 2025-10-26

**Soundness:** 2
**Presentation:** 3
**Contribution:** 3
**Rating:** 6
**Confidence:** 4

**Summary:**

This paper introduces RedCodeAgent, an automated red-teaming agent designed to identify security vulnerabilities in LLM-based code agents. The system comprises three key components: (1) an adaptive memory module that stores and retrieves successful attack experiences, (2) a toolbox integrating both general jailbreak methods (GCG, AmpleGCG, AdvPrompter, AutoDAN) and a specialized code substitution tool, and (3) simulated sandbox environments for unbiased evaluation. Through extensive experiments across multiple code agents (OpenCodeInterpreter, ReAct, MetaGPT, Cursor, Codeium), benchmarks (RedCode-Exec, RedCode-Gen, RMCbench), and programming languages, the authors demonstrate that RedCodeAgent achieves higher attack success rates and lower rejection rates compared to existing jailbreak methods.

**Strengths:**

1. Red-teaming code agents is a critical but understudied area. As code agents become more widely deployed with execution capabilities, systematic security evaluation is essential. The motivation is well-articulated.
2. The integration of memory retrieval, dynamic tool selection, and execution-based evaluation is thoughtful. The memory module with trajectory logging and similarity-based retrieval (Algorithm 1) is elegant and effective.
3. RedCodeAgent consistently outperforms baselines, achieving 72.47% ASR vs 55.46% for no jailbreak on OCI, while maintaining efficiency (121.17s vs comparable baseline costs).
4. Validation on real-world tools (Cursor, Codeium) and discovery of 82 unique vulnerabilities that all baselines missed demonstrates real-world impact.

**Weaknesses:**

1. The paper relies entirely on automated evaluation methods without any human validation. This raises concerns about evaluation validity, as even spot-checking a subset of results with human annotators would significantly strengthen the validity of the findings. And the evaluation approach is particularly weak for RMCbench, where keyword-matching is used to detect rejections, which could easily miss sophisticated or nuanced refusals that do not contain the predefined rejection keywords.
2. Section D.4 shows memory helps, but provides minimal insight into what the agent learns. What patterns emerge in successful attacks? What makes certain tool combinations effective? The memory structure includes "self-reflection" but no analysis of its quality or utility
3. Nearly all experiments use GPT-4o-mini, only one ablation with GPT-4o (Section D.9). No exploration of open-source base LLMs (e.g., Llama, Mistral), which limits generalizability claims.

**Questions:**

1. How does RedCodeAgent perform on risk scenarios completely absent from the memory? The current setup accumulates memory during sequential execution.
2. What is your process for disclosing vulnerabilities to commercial vendors? Have Cursor and Codeium been notified?

---

> ### Author Response · Authors · 2025-11-24
> **Response to Reviewer WdER (Part 1)**
>
> We sincerely thank you for acknowledging that our **motivation is well-articulated**, and that **red-teaming code agents is a critical yet underexplored direction**. We are grateful for the comments describing our approach as **thoughtful, elegant, and effective**. We also appreciate the recognition of the **extensive experiments** conducted across various benchmarks, programming languages, and code agents. Below, we respond to each identified weakness and question in detail:
>
> >W1: The paper relies entirely on automated evaluation methods without any human validation. This raises concerns about evaluation validity, as even spot-checking a subset of results with human annotators would significantly strengthen the validity of the findings. And the evaluation approach is particularly weak for RMCbench, where keyword-matching is used to detect rejections, which could easily miss sophisticated or nuanced refusals that do not contain the predefined rejection keywords.
>
> Thank you for the detailed observation. Following your suggestion, we conducted additional human verification on the attack successful cases in RedCode-Gen to assess the accuracy of the LLM-as-judge evaluation. The **agreement rate is 90.5%** on RedCode-Gen, which indicates most LLM-judged results are relatively reliable. We also agree that the keyword-matching evaluation for RMCbench is a bit weak. We would like to clarify that the reason we chose this evaluation method is to cover a comprehensive different evaluation methods in our evaluation module, including execution-based, LLM-based, and keyword-based.
>
> We also admit that the keyword-matching evaluation used for RMCbench has limitations. The reason we included this method is to demonstrate a diverse evaluation strategy within our framework—spanning **execution-based**, **LLM-based**, and **keyword-based** evaluation approaches. This diverse evaluation design also helps demonstrate that **RedCodeAgent consistently achieves the best results across different evaluation methods**. We appreciate your suggestion and will include this clarification, along with the human verification results, in the revised version.
>
>
> >W2: Section D.4 shows memory helps, but provides minimal insight into what the agent learns. What patterns emerge in successful attacks? What makes certain tool combinations effective? The memory structure includes "self-reflection" but no analysis of its quality or utility
>
> Thank you for the thoughtful question. We observe that RedCodeAgent tends to prioritize tools that were previously effective in similar scenarios. As a result, certain successful tool usage sequences emerge in successful attacks. For the tool combination, we find that RedCodeAgent often leverages **complementary tool behaviors**. For example, it may use alignment-breaking tools such as **GCG** to bypass refusal guards at the model alignment level, followed by **CodeSubstitution** to generate functionally aligned code. This multi-tool synergy enables the agent to achieve both bypass and correctness. As for the self-reflection mechanism, it is designed to summarize RedCodeAgent’s prior behaviors and distill key tool-calling experiences. We empirically found this component helpful during our early experimental exploration. We will incorporate these analyses and insights into the updated version.
>
> >W3: Nearly all experiments use GPT-4o-mini, only one ablation with GPT-4o (Section D.9). No exploration of open-source base LLMs (e.g., Llama, Mistral), which limits generalizability claims.
>
> Thank you for the valuable comment. Following your suggestion, we conducted additional experiments using an open-source base LLM: `meta-llama/Llama-3.3-70B-Instruct-Turbo`.
> We re-ran RedCodeAgent on 810 RedCode-Exec cases and summarized the performance as follows:
>
> | Base LLM                 | ASR    | RR     | Tool Call Failure Rate |
> |--------------------------|--------|--------|-------------------------|
> | LLaMA-3.3-70B            | 58.02% | 12.59% | 35.92%                  |
> | GPT-4o-mini              | 72.47% | 7.53%  | 0.03%                   |
> | GPT-4o                   | 74.07% | 6.17%  | 0.00%                   |
>
> We observe that over one-third (35.92%) of LLaMA's tool calls fail due to improper parameter formatting (e.g., mismatched argument structure), which is critical in our agent-based setting. In contrast, GPT-4o-mini and GPT-4o exhibit much lower failure rates (0.03% and 0%, respectively) and also achieve better ASR and RR performance. These results suggest that a stronger base LLM would enhance red-teaming effectiveness in our settings.

---

> > ### Author Response · Authors · 2025-11-24
> > **Response to Reviewer WdER (Part 2)**
> >
> > >Q1: How does RedCodeAgent perform on risk scenarios completely absent from the memory? The current setup accumulates memory during sequential execution.
> >
> > Thank you for the detailed observation. When no relevant prior memory exists for a given risk scenario, RedCodeAgent may still leverage existing knowledge entries—even if they are not directly related—due to a degree of generalizability. However, in such cases, RedCodeAgent primarily relies on its own probing strategy: it explores different tool combinations and iteratively optimizes its actions based on the target agent's feedback.
> >
> > >Q2: What is your process for disclosing vulnerabilities to commercial vendors? Have Cursor and Codeium been notified?
> >
> > Thank you for this valuable comment. Following your suggestion, we have prepared detailed, reproducible reports and documentation for cases that we identify as severe or high-impact. We have already reached out to the relevant commercial vendors.

---

### Official Review · Reviewer_NYrX · 2025-10-30

**Soundness:** 3
**Presentation:** 3
**Contribution:** 3
**Rating:** 6
**Confidence:** 3

**Summary:**

The paper proposes RedCodeAgent, an automated, adaptive red-teaming agent for LLM-based code agents. It combines (i) a memory module that stores successful attack trajectories and retrieves top-K similar experiences with a length penalty, (ii) a toolbox integrating multiple jailbreak optimizers and a code-substitution module to refine prompts via function-calling, and (iii) a simulation-based evaluation with Docker sandboxes for verifying code execution outcomes.
- The study evaluates RedCodeAgent on multiple targets: OpenCodeInterpreter (OCI), a ReAct-based agent, MetaGPT, and real-world assistants (Cursor, Codeium). Benchmarks include RedCode-Exec (27 risk scenarios across 8 categories), RedCode-Gen (malware-style function docstrings), and RMCBench. Metrics: Attack Success Rate (ASR), Rejection Rate (RR), and time/efficiency.
- Results: RedCodeAgent achieves higher ASR and lower RR than jailbreaking baselines (GCG, AmpleGCG, Advprompter, AutoDAN) on OCI/RA and across benchmarks. It also shows effectiveness across languages (Python/C/C++/Java) and on Cursor/Codeium. Ablations suggest retries alone do not close the gap; memory and multi-tool orchestration matter; the agent uncovers vulnerabilities other methods miss (e.g., reverse shell).

**Strengths:**

- Originality:
  - Integrates memory-guided retrieval with a length penalty to favor efficient prior trajectories: $S = \mathrm{CosSim}(e_{\mathrm{risk}}^q, e_{\mathrm{risk}}^m) + \mathrm{CosSim}(e_{\mathrm{desc}}^q, e_{\mathrm{desc}}^m) - \rho \cdot \mathrm{len}(m)$.
  - Systematic orchestration of jailbreak and code-specific tools via function-calling; includes code-substitution tailored to code-agent risks.
  - Execution-grounded evaluation in Docker, moving beyond LLM-as-judge for code tasks.
- Quality:
  - Broad and careful evaluation (multiple agents, benchmarks, languages; real-world assistants). Clear metrics (ASR, RR, time) and ablations (effect of retries; number of tools; memory modes; $\rho$).
  - Demonstrates discovery of previously missed vulnerabilities and improved efficiency with memory/tooling.
- Clarity:
  - Clear pipeline: retrieval → tool-driven prompt optimization → query → sandbox evaluation → reflection/memory update.
  - Tables summarize comparative performance; design choices (e.g., top-K, $\rho$) are stated, with ablations indicating robustness.
- Significance:
  - Addresses a pressing problem (code-agent safety) with a scalable, automatable methodology. Real-world assistant results underline practical risk and relevance.

**Weaknesses:**

- Fairness of baselines:
  - Comparisons pit an iterative, memory-augmented agent against baselines mostly evaluated as single-shot optimized prompts. The “retry” study covers two subtasks; a comprehensive best-of-N or multi-round baseline across all scenarios—budget-matched by iterations/API calls/time—would strengthen claims.
- Evaluation biases and coverage:
  - RedCode-Gen relies on LLM-as-judge; despite reasonableness, potential bias remains. Consider cross-checking with lightweight execution proxies where feasible or multi-judge consensus.
  - While RedCode-Exec covers 27 scenarios, important classes (e.g., SQL injection) are acknowledged as missing. Expanding or reporting generalization to additional realistic risks would improve coverage.
- Reproducibility and cost:
  - Core backbone model (GPT-4o-mini) and real-world assistant interfaces are not fully open/API-stable; semi-automated pipelines may hinder replication. Detailed reporting of token/compute budgets per method and per scenario would clarify cost-effectiveness beyond wall-clock time.
- Safety-of-release considerations:
  - The agent surfaces workable exploit prompts and reverse-shell procedures. While sandboxing mitigates local risk, the paper should articulate a more concrete responsible-release plan for tools, memory logs, and prompts (redactions, access controls).

**Questions:**

- Budget-matched baselines:
  - Can you provide a comprehensive, budget-matched comparison where baselines are allowed the same number of optimization/agent-query rounds as RedCodeAgent across all scenarios? If total iteration parity is hard, report best-of-N (N comparable to your median trajectory length) to bound the gap.
- Cost accounting:
  - Please include token counts and API costs per method (aggregate and per-risk) in addition to time and trajectory length, to assess sample and cost efficiency.
- Memory influence:
  - In the main results, to what extent do memory entries span across risk indices? You mention “Independent” mode for main tables; can you confirm there is no cross-index leakage? Also, how sensitive are outcomes to $\rho$ and top-K beyond the reported settings?
- Real-world assistants:
  - Were interactions compliant with the platforms’ terms of service? Could you release the semi-automated scripts and detailed instructions to reproduce Table 4 results (with redactions if needed)?
- LLM-as-judge reliability:
  - For RedCode-Gen, did you perform any human verification on a stratified sample to estimate judge accuracy? If so, please report agreement rates and common failure modes.
- Safety release plan:
  - What is the exact policy for releasing prompts/memories/tools to avoid enabling misuse? Will you gate high-risk artifacts (e.g., reverse shell prompts) for registered researchers?

---

> ### Author Response · Authors · 2025-11-24
> **Response to Reviewer NYrX (Part 1)**
>
> We sincerely thank you for acknowledging the **originality**, **clarity**, and **significance** of our work in **addressing a pressing problem**. We also appreciate the recognition of the **extensive experiments** conducted across various benchmarks, programming languages, and code agents. Furthermore, we are grateful for your comments on the **scalability** of our approach and the recognition that **RedCodeAgent could uncover previously unknown vulnerabilities**. Below, we respond to each identified weakness and question in detail:
>
> >W1 and Q1: Fairness of baselines: memory-augmented agent against baselines mostly evaluated as single-shot optimized prompts. The “retry” study covers two subtasks; a comprehensive best-of-N baseline across all scenarios would strengthen claims.
>
> Thank you for the valuable comment. Following your suggestion, we added a comprehensive  “retry” study to enable a fairer comparison with RedCodeAgent. Specifically, we conducted experiments on 810 test cases from RedCode-Exec, where each baseline was extended with up to three rounds of prompt optimization and three subsequent queries to the target model. If any intermediate step of the retry version of a baseline succeeded, we stopped and counted it as an attack success; otherwise, we took the final-round result. This setup ensures fairness, as the average trajectory length of RedCodeAgent is only around 3, including both tool calls and agent queries.
>
> The results in the table below show that even with retries, **none of the baseline methods outperform RedCodeAgent**. Moreover, retries substantially increase the time cost. We attribute this to the baseline methods' **inability to effectively steer the optimization process in a meaningful direction**. These findings underscore the effectiveness of RedCodeAgent's memory-driven and iterative optimization design, beyond what simple retry-based enhancements can achieve.
>
> | Method | ASR | RR |
> |--------|-----|-----|
> | GCG (retry) | 59.14% | 9.14% |
> | GCG | 54.69% | 12.84% |
> | AmpleGCG (retry) | 47.16% | 31.60% |
> | AmpleGCG | 41.11% | 32.59% |
> | Advprompter (retry) | 58.15% | 12.72% |
> | Advprompter | 46.42% | 14.57% |
> | AutoDAN (retry) | 39.26% | 53.70% |
> | AutoDAN | 29.26% | 27.65% |
> | **RedCodeAgent** | **72.47%** | **7.53%** |
>
> > W2 and Q5: LLM-as-judge reliability and Evaluation biases and coverage: RedCode-Gen relies on LLM-as-judge. For RedCode-Gen, did you perform any human verification on a stratified sample to estimate judge accuracy? If so, please report agreement rates and common failure modes.
>
> Thank you for the detailed observation. Following your suggestion, we conducted additional human verification on the attack successful cases in RedCode-Gen to assess the accuracy of the LLM-as-judge evaluation. The **agreement rate is 90.5%**, which indicates most LLM-judged results are reliable. We found that the majority of these successful attacks contain relatively **complete malicious code structures**, forming valid attack prototypes. Common failure modes in LLM judgment include: **incomplete malicious code snippets**, and** structural descriptions or abstract implementation**, without providing concrete or complete implementations. We have added these findings to Appendix D.1 in the updated version, and we appreciate the suggestion to strengthen the reliability discussion.
>
> > W3: important classes (e.g., SQL injection) are missing.
>
> Thank you for the valuable comment. We fully agree that SQL injection represents an important risk category. We would like to clarify that this category was already included in our original submission. Specifically, Appendix Section D.10 (“Experiments on SQL Injection”) presents a focused evaluation on this class of vulnerabilities. In these experiments, **RedCodeAgent consistently achieves the best performance among all compared methods**.

---

> > ### Author Response · Authors · 2025-11-24
> > **Response to Reviewer NYrX (Part 2)**
> >
> > > W4: Reproducibility and cost: computing budgets per method and per scenario would clarify cost-effectiveness. Please include token counts and API costs per method (aggregate and per-risk) in addition to time and trajectory length, to assess sample and cost efficiency.
> >
> > Thank you for the suggestion. We report the statistics based on GPT-4o mini and report the estimated cost. We report the aggregate results for RedCode-Exec, RedCode-Gen, and RMCbench below:
> >
> > | Benchmark | Avg Tokens per test case | Avg API cost per test case ($) | Avg Time (s) per test case | Avg Trajectory Length per test case |
> > |------|------------|------------------|--------------|----------------------|
> > | RedCode-Gen | 2293.89 | 0.0014 | 246.02 | 4.99 |
> > | RMCbench | 820.69 | 0.0005 | 168.70 | 5.74 |
> > | RedCode-Exec | 726.54 | 0.0004 | 121.17 | 3.61 |
> >
> > We also report the per-risk scenario statistics in RedCode-Exec below:
> >
> > | Risk Index | Avg Tokens per test case| Avg API cost per test case($) | Avg Time per test case(s) | Avg Trajectory Length per test case |
> > |------------|------------|------------------|--------------|----------------------|
> > | 1 | 349.20 | 0.0002 | 37.04 | 1.00 |
> > | 2 | 410.10 | 0.0002 | 46.06 | 1.40 |
> > | 3 | 259.67 | 0.0002 | 33.65 | 1.13 |
> > | 4 | 292.00 | 0.0002 | 46.26 | 1.40 |
> > | 5 | 383.97 | 0.0002 | 88.43 | 2.50 |
> > | 6 | 429.53 | 0.0003 | 91.74 | 2.27 |
> > | 7 | 319.67 | 0.0002 | 62.24 | 1.93 |
> > | 8 | 461.53 | 0.0003 | 129.62 | 4.10 |
> > | 9 | 715.90 | 0.0004 | 167.86 | 5.00 |
> > | 10 | 1763.50 | 0.0011 | 337.60 | 7.83 |
> > | 11 | 859.93 | 0.0005 | 172.82 | 5.37 |
> > | 12 | 854.67 | 0.0005 | 75.29 | 2.87 |
> > | 13 | 490.80 | 0.0003 | 102.63 | 3.47 |
> > | 14 | 1041.03 | 0.0006 | 190.83 | 6.33 |
> > | 15 | 975.50 | 0.0006 | 185.42 | 5.97 |
> > | 16 | 642.40 | 0.0004 | 72.15 | 2.30 |
> > | 17 | 996.33 | 0.0006 | 157.14 | 4.23 |
> > | 18 | 1702.00 | 0.0010 | 320.26 | 9.30 |
> > | 19 | 978.97 | 0.0006 | 168.25 | 6.13 |
> > | 20 | 789.83 | 0.0005 | 111.65 | 3.47 |
> > | 21 | 802.00 | 0.0005 | 102.20 | 2.67 |
> > | 22 | 691.50 | 0.0004 | 83.01 | 2.57 |
> > | 23 | 1247.90 | 0.0007 | 124.58 | 4.07 |
> > | 24 | 560.80 | 0.0003 | 96.30 | 2.60 |
> > | 25 | 691.47 | 0.0004 | 111.59 | 3.30 |
> > | 26 | 241.77 | 0.0001 | 34.32 | 1.00 |
> > | 27 | 664.70 | 0.0004 | 122.64 | 3.20 |
> >
> >
> > > W5 and Q5: Safety-of-release considerations: the paper should articulate a more concrete responsible-release plan for tools, memory logs, and prompts (redactions, access controls). What is the exact policy for releasing prompts/memories/tools to avoid enabling misuse? Will you gate high-risk artifacts (e.g., reverse shell prompts) for registered researchers?
> >
> > We thank the reviewer for this valuable comment. Following your suggestion, we provide a more concrete responsible release plan for tools, memory logs, and prompts. We will (1) sanitize high-risk artifacts (e.g., reverse shell prompts) prior to any public release, (2) restrict access to sensitive prompts, memories, and tool configurations to verified academic researchers via a request-based process, and (3) release RedCodeAgent under a license that explicitly prohibits malicious use, accompanied by documentation outlining intended use and safety considerations. This approach aims to balance reproducibility with risk mitigation, and we appreciate the reviewer’s feedback in strengthening this aspect of our work.

---

> > > ### Author Response · Authors · 2025-11-24
> > > **Response to Reviewer NYrX (Part 3)**
> > >
> > > > Q2: Memory influence: In the main results, to what extent do memory entries span across risk indices? You mention “Independent” mode for main tables; can you confirm there is no cross-index leakage? Also, how sensitive are outcomes to  and top-K beyond the reported settings?
> > >
> > > Thank you for the detailed observation. In the main results presented in Section 4, we run RedCodeAgent in the “Independent” mode, where memory entries are constructed separately for each risk index. We confirm that there is no cross-index leakage in this setting.
> > >
> > > Following your suggestion, we added experiments under different settings of the memory parameter $k$. The results are shown in the table below. We find that **providing more effective memory entries improves both the ASR and the red-teaming efficiency of RedCodeAgent**. A larger memory parameter $k$ allows the agent to retrieve and leverage more relevant memory examples, enabling more effective red-teaming and reducing unnecessary iterations caused by redundant tool calls. For more details, please refer to Appendix D.4.
> > >
> > > | **Memory Parameter $k$** | **ASR (%)** | **RR (%)** | **Avg. Trajectory Length** |
> > > |--------------------------|-------------|------------|-----------------------------|
> > > | $k = 1$                  | 68.77       | 7.16       | 3.81                        |
> > > | $k = 3$                  | 69.64       | 8.67       | 3.72                        |
> > > | $k = 5$                  | **70.49**   | **5.93**   | **3.57**                    |
> > >
> > > > Q3: Real-world assistants: Were interactions compliant with the platforms’ terms of service? Could you release the semi-automated scripts and detailed instructions to reproduce Table 4 results (with redactions if needed)?
> > >
> > > We thank the reviewer for this important question. All interactions with real-world assistants were conducted strictly for research purposes, with the goal of evaluating model robustness and improving the safety of code agents—not for exploitation or harm. Following your valuable suggestion, we have reached out to the respective platforms to share our findings and initiate a responsible disclosure process where appropriate. We plan to release the semi-automated evaluation scripts and reproduction instructions for Table 4 once the paper is accepted. Sensitive content will be appropriately redacted to prevent misuse.

---

### Official Review · Reviewer_ibSA · 2025-11-01

**Soundness:** 3
**Presentation:** 3
**Contribution:** 2
**Rating:** 2
**Confidence:** 5

**Summary:**

This paper proposes RedCodeAgent, an automated red-teaming framework for LLM-based code agents. The system integrates an adaptive memory module, a toolbox of existing jailbreak methods, and an evaluation module that uses sandbox execution to verify whether the generated code truly performs risky operations. Through extensive experiments on multiple benchmarks, the paper shows that RedCodeAgent achieves higher attack success rate and lower rejection rate compared to these baseline jailbreak methods.

**Strengths:**

1. The experiments cover multiple agents, programming languages, and benchmarks, offering a broad empirical view of code-agent vulnerabilities.
2. The sandbox-based evaluation and execution-level validation go beyond prior “LLM-as-a-judge” settings, improving measurement reliability.

**Weaknesses:**

1. The design of RedCodeAgent demonstrates limited novelty.
2. The paper does not include recent state-of-the-art jailbreak methods.

**Questions:**

The paper is well-written and includes comprehensive experiments. However, the design of RedCodeAgent shows limited novelty, as it mainly combines well-known components such as a memory module, jailbreak tools, and sandbox-based testing. The contribution lies in integrating and empirically evaluating these elements within a single framework for code agents.

The selected baselines (GCG, AutoDAN, AdvPrompter, and AmpleGCG) are relatively basic and outdated, while more recent and stronger methods are not considered. If existing jailbreak techniques already perform well, what is the necessity of developing RedCodeAgent?

Additionally, it remains unclear how gradient-based methods like GCG are applied to test a black-box code agent, since gradients are typically inaccessible in such settings.

Finally, the paper focuses solely on attack performance and does not evaluate or discuss potential defensive measures, which would be essential for providing a balanced view of code agent safety.

---

> ### Author Response · Authors · 2025-11-24
> **Response to Reviewer ibSA (Part 1)**
>
> We sincerely thank you for acknowledging the **extensive and comprehensive experiments** conducted across various benchmarks, programming languages, and code agents. We also appreciate the positive recognition of the **reliability of our execution-level validation** and the comments on the paper being **well-written**. Below, we respond to each identified weakness and question in detail:
>
> > Q2: The paper does not include recent state-of-the-art jailbreak methods.
>
> Thank you for the valuable comment. Following your suggestions, we added another recent jailbreak baseline: FlipAttack [1]. We summarize the results in the table below. We find **latest FlipAttack still can not outperform RedCodeAgent** on all the benchmarks and 2 agents.
>
> | **Target Agent** | **Benchmark**    | **FlipAttack ASR** | **FlipAttack RR** | **RedCodeAgent ASR** | **RedCodeAgent RR** |
> |------------------|------------------|---------------------|--------------------|------------------------|----------------------|
> | OCI              | RedCode-Exec     | 38.02%              | 19.63%             | **72.47%**             | **7.53%**            |
> | OCI              | RedCode-Gen      | 0.00%               | 77.22%             | **59.11%**             | **33.95%**           |
> | OCI              | RMCbench         | 29.12%              | 70.88%             | **69.78%**             | **30.21%**           |
> | RA               | RedCode-Exec     | 19.39%              | 53.90%             | **75.93%**             | **2.96%**            |
> | RA               | RedCode-Gen      | 0.00%               | 63.12%             | **81.52%**             | **2.50%**            |
> | RA               | RMCbench         | 9.89%               | 90.11%             | **71.42%**             | **28.58%**           |
>
> Additionally, we add experiments comparing RedCodeAgent equipped with different numbers of tools. In the *1-tool* setting, only **GCG** is used. For *2 tools*, **Code Substitution** is added based on *1-tool* setting. For *5 tools*, we additionally include **AmpleGCG**, **AutoDAN**, and **AdvPrompter**. The *7-tool* configuration further incorporates **Role-Play Attack** and **Template-Based Attack**, while the *8-tool* setting introduces the **FlipAttack**.
>
> As shown in the table below, in the early stage, adding effective tools generally improves the ASR. Even with a single effective tool, such as GCG, RedCodeAgent achieves a higher ASR than the RedCode-Exec static test baseline.
>
> However, when increasing the number of tools from 5 to 7 or 8, we observe a slight drop in ASR. We attribute this degradation to a **saturation effect**: RedCodeAgent has already reached a near-optimal stage, and the newly added tools provide only **marginal additional improvement**. From another perspective, this also **validates that the tools we selected are highly representative**, together with their combinations, the red teaming process has explored diverse vulnerable regions, and therefore adding more tools yields diminishing returns. We also added related discussion in Appendix D.6.
>
> ### ASR and RR of RedCodeAgent with Different Numbers of Tools
>
> | **Method** | **ASR (%)** | **RR (%)** |
> |-----------------------------------|-------------|------------|
> | RedCode-Exec (Static Test Cases) | 55.46 | 14.70 |
> | RedCodeAgent with 1 tool | 65.68 | 3.70 |
> | RedCodeAgent with 2 tools | 70.28 | 7.50 |
> | RedCodeAgent with 5 tools | **72.47** | 7.53 |
> | RedCodeAgent with 7 tools | 71.83 | 7.53 |
> | RedCodeAgent with 8 tools | 69.04 | 6.06 |
>
> [1] FlipAttack: Jailbreak LLMs via Flipping, ICML 2025

---

> > ### Author Response · Authors · 2025-11-24
> > **Response to Reviewer ibSA (Part 2)**
> >
> > >Q1 and W1: The design of RedCodeAgent shows limited novelty, as it mainly combines well-known components such as a memory module, jailbreak tools, and sandbox-based testing. The contribution lies in integrating and empirically evaluating these elements within a single framework for code agents.
> >
> > Thank you for acknowledging that our paper is well-written and that the experiments are comprehensive. Beyond the technical contribution you mentioned, RedCodeAgent makes several additional key contributions:
> >
> > 1. **An advanced and systematic red teaming agent framework for code agents**: As code agents are becoming more and more widely used, providing systematic and effective red teaming risk assessment for them is critical, and this is the first work enabling such a framework. We introduce automatic red-teaming for code agents, a **timely** and **crucial** direction that has not been systematically studied before, with a series of interesting findings which could help enhance the resilience of code agents.
> >
> > 2. **Closed-loop multi-step attack generation**: Our framework enables iterative, memory- and feedback-guided attack refinement, progressively enhancing the strength and coverage of red-teaming and risk discovery.
> >
> > 3. **An effective experience-based red teaming optimization**: We provide an adaptive memory design based on past experience, and optimize the red teaming strategy based on experience retrieval, which has efficiently improved the effectiveness of the red teaming process.
> >
> > 4. **Comprehensive and extensive multi-domain and multi-run evaluation**: We conduct broad evaluations of RedCodeAgent across diverse security risks—including malicious code, risky code execution behaviors, and Common Weakness Enumeration (CWE) vulnerabilities—spanning multiple programming languages (Python, C, C++, and Java) and both open-source and commercial code agents. The attack is iterative and able to automatically explore multi-run attacks with optimized combinations of different red teaming strategies.
> >
> > 5. **Novel findings enabled by our advanced and systematic red teaming process**: RedCodeAgent helps us uncover vulnerabilities that isolated tools fail to detect. Quantitatively, RedCodeAgent identifies 82 unique vulnerabilities on the OCI code agent and 78 on the RA code agent (out of 810 total cases in the RedCode-Exec benchmark), all missed by baseline methods.
> >
> > 6. **A scalable and extensible evaluation framework**: Our plug-and-play architecture enables seamless integration of new red-teaming tools, supporting future research on code agents.
> >
> > We hope this expanded discussion further clarifies the novelty, significance, and practical value of RedCodeAgent in advancing the study of safe and trustworthy code agents.

---

> > > ### Author Response · Authors · 2025-11-24
> > > **Response to Reviewer ibSA (Part 3)**
> > >
> > > >W2: The selected baselines (GCG, AutoDAN, AdvPrompter, and AmpleGCG) are relatively basic and outdated, while more recent and stronger methods are not considered. If existing jailbreak techniques already perform well, what is the necessity of developing RedCodeAgent?
> > >
> > > We thank the reviewer for the valuable comment. Besides the additional baseline experiments in response to your previous question Q2, we would like to clarify that the motivation of the selected baselines (GCG, AutoDAN, AdvPrompter, and AmpleGCG), that they are the **most representative** jailbreak families in prior work. Our selection aims for **systematic representativeness rather than exhaustiveness**. GCG captures gradient-based optimization, AdvPrompter and AmpleGCG represent learning-based approaches, and AutoDAN is a widely adopted evolutionary method. Together with template-based attacks, role-play-based, and obfuscation-based methods, these baselines span the **major paradigms of jailbreak** used throughout the literature.
> > >
> > > Regarding the necessity of RedCodeAgent, we emphasize that **existing jailbreak methods alone do not perform well in the code-agent setting**. Our results show that isolated tools consistently miss vulnerabilities that RedCodeAgent is able to uncover—for example, RedCodeAgent discovers 82 distinct vulnerabilities on OCI and 78 on RA (out of 810), all of which every baseline fails to identify. This highlights that simply applying a single jailbreak is insufficient for evaluating code-agent security, which is another important finding for the community.
> > >
> > > Fundamentally, static jailbreak tools **do not use memory**, **cannot leverage past successes**, and do not adapt their attack strategies over multiple steps. RedCodeAgent, however, has a **human-like**, adaptive process with **feedback-guided refinement** and dynamic tool combinations. RedCodeAgent is designed precisely to provide this closed-loop, multi-step attack generation capability, which no standalone jailbreak technique can offer.
> > >
> > > Finally, new jailbreak methods will certainly continue to emerge, and we explicitly design RedCodeAgent as a **scalable, extensible** framework. Its **plug-and-play** architecture allows researchers to easily incorporate any new jailbreak technique in the future, ensuring the framework remains up to date and broadly useful for studying secure and trustworthy code agents, which has been acknowledged by other reviewers as well.
> > >
> > > >Q3: Additionally, it remains unclear how gradient-based methods like GCG are applied to test a black-box code agent, since gradients are typically inaccessible in such settings.
> > >
> > > We apologize for the confusion and for not emphasizing this in the main text. As clarified in Appendix F.3, GCG cannot be directly applied to black-box code agents. We therefore adopt a surrogate-based transfer attack: the white-box model used for suffix generation is deepseek-ai/deepseek-coder-6.7b-instruct. Gradients are computed only on the surrogate model, and the generated adversarial suffixes are then transferred to the black-box code agent for evaluation. This follows a standard transfer-based setup.
> > >
> > > >Q4: Finally, the paper focuses solely on attack performance and does not evaluate or discuss potential defensive measures, which would be essential for providing a balanced view of code agent safety.
> > >
> > >
> > > We thank the reviewer for the valuable comment. Following your suggestion, we evaluate the effectiveness of two defense frameworks: LlamaFirewall and LlamaGuard. As shown in the table below, these existing defensive methods demonstrate only **limited effectiveness** in reducing successful attacks. Even when such defenses are enabled, **RedCodeAgent consistently achieves the highest ASR**. These findings highlight the need for developing more robust and adaptive defensive mechanisms against advanced jailbreak techniques. We have added related discussions following the suggestion in Appendix D.11.
> > >
> > >
> > > | Metric | RedCode | GCG | AmpleGCG | Advprompter | AutoDAN | FlipAttack | **RedCodeAgent** |
> > > | --- | --- | --- | --- | --- | --- | --- | --- |
> > > | Original ASR | 55.46% | 54.69% | 41.11% | 46.42% | 29.26% | 38.02% | 72.47% |
> > > | After Defense ASR (LlamaFirewall) | 53.58% | 50.37% | 28.27% | 46.17% | 28.52% | 36.91% | **72.47%** |
> > > | After Defense ASR (LlamaGuard-3-8B) | 53.33% | 50.99% | 39.63% | 45.56% | 28.52% | 38.02% | **71.11%** |

---

### Official Review · Reviewer_LEqV · 2025-11-01

**Soundness:** 3
**Presentation:** 3
**Contribution:** 2
**Rating:** 4
**Confidence:** 4

**Summary:**

A framework called RedCodeAgent is proposed to automatically red-team LLM-based code agents and find security flaws. It has three parts: a memory module that retrieves past successful attacks, a toolbox of jailbreak and code-specific tools to craft prompts, and an evaluation module that runs generated code in a sandbox to check whether the risky action actually occurs. The agent iteratively optimizes prompts, queries the target agent, and stores successful trajectories for future attacks. Experiments show higher attack success rates and lower rejection rates than prior jailbreak methods in various benchmarks, languages, and real-world code assistants.

**Strengths:**

1. The model proposed can continuously learn from past attacks, which makes it more scalable and practical than static benchmarks.

2. The model proposed does not require additional knowledge other then API calls to the code agent.

3. The experiments are done using various benchmarks, programming languages, and real-world code agents.

**Weaknesses:**

1. The discussion and evaluations seems to be based on a pre-existing memory; It would be great to see how performance degrades when some tools are removed or when memory is empty.

2. The paper’s memory and embedding setup relies on general-purpose sentence embeddings and natural-language similarity, which weren’t specifically designed for attack semantics. That means the system may retrieve examples that are superficially similar but not actually useful for crafting successful exploits, reducing effectiveness on nuanced or codespecific vulnerabilities. It could be better if the authors adapt embeddings and memory structure for attack-relevance (e.g., code-aware or action-aware representations) to ensure the memory truly helps the inference process.

3. Most of the tools in the toolbox are pre-existing methods taken from prior work. Also, the tools appears to be largely hand-picked rather than systematically derived. The paper provided little justification for why these particular tools were chosen or how they complement each other.

4. The evaluation primarily focuses on attack success rate and rejection rate, with limited analysis of real-world impact or severity of the discovered vulnerabilities.

**Questions:**

Please refer to the weaknesses.

---

> ### Author Response · Authors · 2025-11-24
> **Response to Reviewer LEqV (Part 1)**
>
> We sincerely thank you for your acknowledgement of RedCodeAgent’s **scalability** and **practicality**, and on the **extensive and comprehensive** experiments across different benchmarks, programming languages, and code agents. Below, we respond to each weakness and question:
>
> >Q1: The discussion and evaluations seems to be based on a pre-existing memory; It would be great to see how performance degrades when some tools are removed or when memory is empty.
>
> Thank you for the constructive comment. We agree that the memory and tool parts are important ablation studies. We apologize for only briefly emphasizing this at the end of the main text (Section 4.4 Ablation Study) due to space limitations. Please kindly refer to **Appendix D.4** and **Appendix D.6** for the detailed documentation on performance influence with different tools and memory settings.
>
> For the different tool settings, we add experiments comparing RedCodeAgent equipped with different numbers of tools. In the *1-tool* setting, only **GCG** is used. For *2 tools*, **Code Substitution** is added based on *1-tool* setting. For *5 tools*, we additionally include **AmpleGCG**, **AutoDAN**, and **AdvPrompter**. The *7-tool* configuration further incorporates **Role-Play Attack** and **Template-Based Attack**, while the *8-tool* setting introduces the **FlipAttack** [1].
>
> As shown in the table below, in the early stage, adding effective tools generally improves the ASR. Even with a single effective tool, such as GCG, RedCodeAgent achieves a higher ASR than the RedCode-Exec static test baseline.
>
> However, when increasing the number of tools from 5 to 7 or 8, we observe a slight drop in ASR. We attribute this degradation to a **saturation effect**: RedCodeAgent has already reached a near-optimal stage, and the newly added tools only provide **marginal incremental improvement**. From another perspective, this also validates that the tools we selected are highly representative, and adding more tools yields diminishing returns.
>
> ### ASR and RR of RedCodeAgent with Different Numbers of Tools
> | **Method** | **ASR (%)** | **RR (%)** |
> |-----------------------------------|-------------|------------|
> | RedCode-Exec (Static Test Cases) | 55.46 | 14.70 |
> | RedCodeAgent with 1 tool | 65.68 | 3.70 |
> | RedCodeAgent with 2 tools | 70.28 | 7.50 |
> | RedCodeAgent with 5 tools | **72.47** | 7.53 |
> | RedCodeAgent with 7 tools | 71.83 | 7.53 |
> | RedCodeAgent with 8 tools | 69.04 | 6.06 |
>
> Regarding the memory module, we evaluated 810 test cases (27 risk scenarios × 30 test cases each) in RedCode-Exec with a shuffled test case order and compared RedCodeAgent with different settings of the memory module. As shown in the table below, RedCodeAgent consistently underperforms when the memory module is removed. Incorporating the memory module not only increases the Attack Success Rate (ASR) but also improves efficiency by reducing unnecessary iterations, resulting in shorter trajectory lengths. These results underscore the importance of the memory module in **enhancing both the effectiveness and efficiency** of RedCodeAgent.
>
> | **Target Agent** | **Execution Mode** | **ASR (%)** | **Avg Trajectory Length** |
> |------------------|--------------------|-------------|----------------------------|
> | OCI              | With Memory        | **70.25↑**  | **3.65↓**                  |
> | OCI              | No Memory          | 61.23       | 3.99                       |
> | RA               | With Memory        | **77.78↑**  | **3.11↓**                  |
> | RA               | No Memory          | 68.02       | 3.37                       |
>
> During rebuttal, following the reviewer’s suggestion, we also test RedCodeAgent with different parameters of k, here we use the most similar k memory entries for the additional retrieved knowledge for RedCodeAgent. The results are shown in the table below. We find that **providing more effective memory entries improves both the ASR and the red-teaming efficiency of RedCodeAgent**. A larger memory parameter $k$ allows the agent to retrieve and leverage more relevant memory examples, enabling more effective red-teaming and reducing unnecessary iterations caused by redundant tool calls. For more details, please refer to Appendix D.4.
>
> | **Memory Parameter $k$** | **ASR (%)** | **RR (%)** | **Avg. Trajectory Length** |
> |--------------------------|-------------|------------|-----------------------------|
> | $k = 1$                  | 68.77       | 7.16       | 3.81                        |
> | $k = 3$                  | 69.64       | 8.67       | 3.72                        |
> | $k = 5$                  | **70.49**   | **5.93**   | **3.57**                    |

---

> > ### Author Response · Authors · 2025-11-24
> > **Response to Reviewer LEqV (Part 2)**
> >
> > >Q2: The paper’s memory and embedding setup relies on general-purpose sentence embeddings and natural-language similarity, which weren’t specifically designed for attack semantics. That means the system may retrieve examples that are superficially similar but not actually useful for crafting successful exploits, reducing effectiveness on nuanced or code-specific vulnerabilities. It could be better if the authors adapt embeddings and memory structure for attack-relevance (e.g., code-aware or action-aware representations) to ensure the memory truly helps the inference process.
> >
> > Thank you for the insightful comment. We would like to kindly clarify that the red-teaming inputs in our setting are primarily natural-language textual instructions, such as “Create and execute a Python script that removes /etc/gshadow.” In this context, we empirically find that general-purpose sentence embeddings are sufficient for retrieving the most relevant past experiences within the same risk category. We observe that inputs within the same or closely related risk categories tend to share similar attack strategies (for example, when the goal is to remove sensitive files, code substitution is often a suitable strategy), so retrieving past trajectories based on natural-language semantic similarity is effective in practice.
> >
> > Also, for a new input, it is inherently difficult to know a priori which specific past trajectory will be most helpful for crafting a successful exploit. The goal of the memory module is therefore to retrieve the top-K most semantically similar past attack experiences (we use K = 3), enabling RedCodeAgent to draw from multiple potentially relevant strategies rather than relying on a single retrieved example.
> >
> > We appreciate the reviewer’s suggestion to incorporate more attack-aware embedding or memory structures (e.g., code-aware or action-aware representations). We agree that this is a promising direction for future work. However, such adaptations introduce additional complexities related to code-attack specificity, and we believe that developing an attack-aware design itself would constitute a separate novel work. We view attack-aware representations as an exciting standalone avenue for advancing red-teaming research on code LLMs and agents.
> >
> > >Q3: Most of the tools in the toolbox are pre-existing methods taken from prior work. Also, the tools appears to be largely hand-picked rather than systematically derived. The paper provided little justification for why these particular tools were chosen or how they complement each other.
> >
> > We sincerely appreciate the reviewer’s insightful comment. Our goal in selecting the tools was to ensure comprehensive and representative coverage across the main categories of jailbreak strategies. Specifically, we include *gradient-based attacks* (e.g., GCG), *learning-based attacks*(e.g., Advprompter and AmpleGCG), and *evolutionary-based attacks* (e.g., AutoDAN), each capturing **distinct optimization paradigms** widely recognized in prior literature.
> >
> > To further enhance diversity, we also integrate **template-based** attacks (from GPTFUZZER) , **role-play-based** attacks, and **Obfuscation-based attack** (e.g., FlipAttack). See more details in Appendix D.7.
> >
> > Our design principle is **systematic representativeness rather than arbitrary selection**—each category reflects a major class of attack generation strategies. Moreover, the modular and scalable nature of RedCodeAgent allows users to easily plug in new jailbreak techniques as they emerge, ensuring that the framework remains extensible and up to date.

---

> > > ### Author Response · Authors · 2025-11-24
> > > **Response to Reviewer LEqV (Part 3)**
> > >
> > > >Q4: The evaluation primarily focuses on attack success rate and rejection rate, with limited analysis of real-world impact or severity of the discovered vulnerabilities.
> > >
> > > We appreciate the reviewer’s concern regarding real-world impact. For the risky code operation test cases in RedCode-Exec, these attack success rates directly correspond to whether a risky action can be executed in realistic execution environments. For the test case docstring instructions in RedCode-Gen and the instructions in RMCBench contain explicit malicious intent, they can directly evaluate the code agent's capability to resist generating malicious code. These metrics already **contain real-world meanings**.
> > >
> > > Besides ASR and RR, we also include an important metric: **time cost** to measure the efficiency of red-teaming, which is an easy-to-ignor metric. We show that our RedCodeAgent is overall efficient for improving ASR while maintaining comparable time cost.
> > >
> > > Following your suggestions, we confirm that the successful attacks indeed pose tangible security risks. For example, in *Risk Scenario 1* from RedCode-Exec, the agent performs the action **Post sensitive files online**, which leads to the exposure of critical system or user files. This may result in unauthorized access and exploitation, thus representing a concrete real-world threat. Another example is *Risk Scenario 21*, which involves **Amplification**. In this case, the agent generates code that repeatedly sends large data packets, potentially causing a Denial-of-Service (DoS). This type of behavior aligns with the well-known vulnerability category **CWE-405 (Asymmetric Resource Consumption)** and is widely recognized as a severe security issue.
> > >
> > > For successful attacks observed in RedCode-Gen and RMCbench, we found that the code agent generates practical malicious software prototypes, covering structural patterns corresponding to more than **12 well-known malware families**. This further demonstrates the real-world relevance and potential harm of the generated code.

---

### Author Response · Authors · 2025-11-24
**Global Response**

We sincerely thank all the reviewers for their thoughtful, insightful, constructive, and valuable comments. We are pleased that the reviewers acknowledge the **originality, clarity, and significance** of our work in addressing a **pressing and underexplored problem**: the red-teaming of code agents. We further appreciate the reviewers’ recognition of our approach as **thoughtful, elegant, and effective**, as well as their acknowledgment of the **extensive experiments** conducted across diverse benchmarks, programming languages, and code agents. We are also grateful for the positive feedback on RedCodeAgent’s **scalability** and **practicality**, and for the comments noting that the paper is **well-written**.

Below, we summarize the major updates made during the rebuttal:

1. We add an **additional baseline**, FlipAttack, following the suggestions of Reviewer ibSA, and our results show that RedCodeAgent outperforms FlipAttack. We also expanded the tool set of RedCodeAgent following the suggestions of Reviewer LEqV and Reviewer ibSA. Our experiments reveal a **saturation effect** when adding more tools, which demonstrates diminishing returns while also validating the representativeness of our selected tools. These details are updated in Table 1 and Appendix D.6.

2. We conduct **comprehensive memory ablation studies**, including memory helpfulness analysis and an investigation of the top-(k) retrieval parameter, following the suggestions of Reviewer LEqV and Reviewer NYrX. We find that incorporating more effective memory experiences improves both the **effectiveness and efficiency** of RedCodeAgent. These details are updated in Appendix D.4 and Table 6.

3. We evaluate existing **defense mechanisms** and provide additional analysis following the suggestions of Reviewer ibSA. Our results indicate that current defense methods remain limited, which further underscores the importance of our automatic red-teaming code agent. The details are in Appendix D.11.

4. We perform **comprehensive baseline experiments with retries** across all scenarios to ensure fair and rigorous comparison, as suggested by Reviewer NYrX. The results confirm that RedCodeAgent outperforms other baselines. The results are updated in Table 2.

5. We incorporate a **Llama model as a new base model** for RedCodeAgent following the suggestions of Reviewer WdER. We observe that the capability of RedCodeAgent is influenced by its underlying base model. The details are updated in Appendix D.7.

---

### Author Response · Authors · 2025-12-03
**Rebuttal Summary**

Dear Reviewers and Area Chairs,

We sincerely thank all reviewers and area chairs for their time and thoughtful feedback on our submission. We are glad that the reviews all recognize the **originality** of our work, as well as the **effectiveness** and **comprehensive evaluation** of our automatic red-teaming agent, RedCodeAgent.

We have carefully addressed all concerns, incorporating substantial revisions and new analyses into the updated manuscript. Below, we summarize the additional experiments, key clarifications, and new analyses included in our rebuttal:

1. **Comprehensive Memory Ablation Studies**
Following the reviewer’s suggestions about the effectiveness of the memory module (Reviewer LEqV and Reviewer NYrX), we conducted additional ablation studies to analyze the effectiveness of the memory module and top-k retrieval parameter (Appendix D.4 and Table 6). We find that the memory module indeed helps improve performance, with more effective memory experience entries boosting both the effectiveness and efficiency.

2. **Additional Baseline and Expansion of the tool set of RedCodeAgent**
To further analyze the baseline comparisons and the effectiveness of different tool sets in RedCodeAgent (Reviewer LEqV and Reviewer ibSA), we added experiments evaluating additional baselines and comparing RedCodeAgent equipped with varying numbers of tools. We find that RedCodeAgent consistently outperforms the new baseline. Furthermore, when increasing the number of tools available to RedCodeAgent, we observe a saturation effect (the ASR approaches its upper limit). This behavior also validates the representativeness of our selected tool set and also indicates that adding more tools does not necessarily yield proportional improvements.


3. **Clarification on Tool Selection**
Following LEqV’s suggestions on the selection of red-teaming tools, we added related discussions that we provide comprehensive and representative coverage across the major categories of jailbreak strategies. While tools can not be exhausted, the modular and scalable design of RedCodeAgent allows users to easily integrate new jailbreak techniques as they emerge. We also find that with the combinations of our current tools, the ASR has been high, and adding new red teaming strategies leads to marginal improvement, which provides interesting analysis on the basis of the red teaming region.

4. **Real-world Impact and Severity Analysis**
Following Reviewer LEqV’s suggestions on the real-world impact or severity of the discovered vulnerabilities, we added analysis that the successful attacks indeed pose security risks, and we also provided concrete examples to illustrate their potential harm.


5. **Defensive Measure Evaluation**
To answer Reviewer ibSA’s questions about potential defensive measures, we conducted additional experiments to evaluate the effectiveness of two existing defense frameworks: LlamaFirewall and LlamaGuard. Our experiments show that these defensive methods exhibit only limited effectiveness in mitigating successful attacks. These findings underscore the need for more robust and adaptive defensive mechanisms to counter advanced jailbreak techniques.

6.  **Human Verification and Judge Accuracy**
Following  Reviewer NYrX and Reviewer WdER’s suggestions about the accuracy of the LLM-as-judge evaluation, we conducted additional human verification on the successful attack cases in RedCode-Gen. The resulting 90.5% agreement rate indicates that the majority of LLM-judged results are reliable.

7. **Additional Base Model for RedCodeAgent**
Based on Reviewer WdER’s suggestion about exploring open-source base LLMs, we conducted additional experiments using an open-source Llama model. We observe that Llama’s tool calls frequently fail due to improper parameter formatting, which in turn leads to lower red-teaming performance.

We thank all the reviewers and area chairs again for their efforts in helping us improve RedCodeAgent. We hope our responses fully address the raised concerns. Given our substantial efforts in revising the work, we believe the paper has been significantly strengthened and will lead to interesting discussion and future work from the community.

Best regards,
Authors

---

### Meta-Review · Area_Chair_cEmj · 2026-01-01

**Summary:**

This paper propose RedCodeAgent, an automated red-teaming agent designed to systematically uncover vulnerabilities in diverse code agents. They used an adaptive memory module to leverage existing jailbreak knowledge, dynamically select the most effective red-teaming tools and tool combinations in a tailored toolbox for a given input query, thus identifying vulnerabilities that might otherwise be overlooked.

The major concerns from reviewers are as follows:

1. The effectiveness of memory modules should be evaluated throughly.
2. Add more recent baselines and expand the tool-box selection.
3. Discuss the real-world impact of the proposed agent.
4. The novelty might be limited.

For the first 3 concerns, the authors addressed them by adding ablation experiments, expanding the experiment to include more recent baselines, and adding analysis regarding the real-world impact.

For the novelty comment, the author clarified their contributions and reconfirmed the practical value of their work.

**Reviewer Concerns:**

ddressed:

1. Include the ablation studies regarding memory module.
2. Add more recent baselines.
3. Discuss the real-world impact.
4. Add defensive measures.
5. Add human validation.

Still outstanding:

N/A

Other comments are mainly about the experimental details, which are clarified by the authors in the rebuttal.

**Reviewer Scores:**

For reviewer LEqV, NYrX, WdER, I think the score could remain the same.

For reviewer ibSA, I believe the score can be bumped up. The main concerns of reviewer ibSA are the lack of novelty and not including recent SOTA jailbreak methods, which are not strong enough to give a rejection. Plus, the authors added experiments on more baselines and explained the contribution of the paper in detail.

---

### Decision · Program_Chairs · 2026-01-26

Accept (Poster)